# Diagnostic Performance of PET or PET/CT Using ^18^F-FDG Labeled White Blood Cells in Infectious Diseases: A Systematic Review and a Bivariate Meta-Analysis

**DOI:** 10.3390/diagnostics9020060

**Published:** 2019-06-15

**Authors:** Marie Meyer, Nathalie Testart, Mario Jreige, Christel Kamani, Mohammed Moshebah, Barbara Muoio, Marie Nicod-Lalonde, Niklaus Schaefer, Luca Giovanella, John O. Prior, Giorgio Treglia

**Affiliations:** 1Department of Nuclear Medicine and Molecular Imaging, Lausanne University Hospital and University of Lausanne, CH-1011 Lausanne, Switzerland; Marie-Madeleine.Meyer@chuv.ch (M.M.); Nathalie.Testart@chuv.ch (N.T.); Mario.Jreige@chuv.ch (M.J.); Christel-Hermann.Kamani@chuv.ch (C.K.); Mohammed.Moshebah@chuv.ch (M.M.); Marie.Nicod-Lalonde@chuv.ch (M.N.-L.); Niklaus.Schaefer@chuv.ch (N.S.); John.Prior@chuv.ch (J.O.P.); 2Department of Internal Medicine, Ente Ospedaliero Cantonale, CH-6500 Bellinzona, Switzerland; Barbara.Muoio@eoc.ch; 3Clinic of Nuclear Medicine and Molecular Imaging, Imaging Institute of Southern Switzerland, Ente Ospedaliero Cantonale, CH-6500 Bellinzona, Switzerland; Luca.Giovanella@eoc.ch; 4Health Technology Assessment Unit, Ente Ospedaliero Cantonale, CH-6500 Bellinzona, Switzerland

**Keywords:** PET, infection, ^18^F-FDG, leukocytes, white blood cells, meta-analysis

## Abstract

Background: Diagnostic performance of positron emission tomography using white blood cells labeled with fluorine-18-fluorodeoxyglucose (^18^F-FDG-WBC PET or PET/CT) in patients with suspicious infectious diseases has been evaluated in several studies; however, there is no consensus about the diagnostic accuracy of this method. Therefore, a systematic review and meta-analysis was carried out on this topic. Methods: A comprehensive computer literature search screening PubMed/MEDLINE, Embase and Cochrane library databases through March 2019 was performed. Pooled sensitivity, specificity, positive and negative likelihood ratios (LR+ and LR−), and diagnostic odds ratio (DOR) of ^18^F-FDG-WBC PET or PET/CT in patients with infectious diseases were calculated. Results: Eight studies on the use of ^18^F-FDG-WBC PET or PET/CT in suspicious infectious diseases were discussed in the systematic review. The meta-analysis of seven studies (236 patients) provided these pooled results on a per patient-based analysis: sensitivity was 86.3% [95% confidence interval (95%CI) 75–92.9%], specificity 92% (95%CI 79.8–97.1%), LR+ 6.6 (95%CI: 3.1–14.1), LR− 0.2 (95%CI: 0.12–0.33), DOR 43.5 (95%CI: 12.2–155). A statistically significant heterogeneity was not detected. Conclusions: Despite limited literature data, ^18^F-FDG-WBC PET or PET/CT demonstrated a good diagnostic accuracy for the diagnosis of infectious diseases; nevertheless, larger studies are needed.

## 1. Introduction

Infectious diseases are a frequent cause of morbidity and mortality worldwide [1]. Early and accurate diagnosis of infectious diseases can be difficult and time-consuming, whereas a delayed diagnosis can be life-threatening. Accurate and early detection and localization of infectious diseases is crucial for patient management and treatment, as well as for the cost containment [2,3].

Clinical history, physical examination and various laboratory tests are performed in patients with suspicious infection [2,3]. Several imaging methods can be used for localization of infectious foci. The recent development of molecular imaging methods has enabled an improved diagnosis of infectious diseases: Molecular imaging methods may detect infection and inflammation in an early phase before the appearance of morphological changes [3,4,5,6]. To this regard, several radiolabeled molecular probes are currently available for imaging of infectious diseases, including both non-specific agents as fluorine-18-fluorodeoxyglucose (^18^F-FDG), a radiolabeled glucose analogue that accumulates similarly in infection, sterile inflammation, and tumor lesions, and more targeted probes that seek to differentiate infection from sterile inflammation [3,4,5,6].

Hybrid ^18^F-FDG positron emission tomography/computed tomography (PET/CT) may be used to evaluate patients with suspicious infection and inflammation, due to the increased ^18^F-FDG uptake in inflammatory cells, based on their high glycolytic metabolism. This imaging technique can identify the source of infection or inflammation before morphological changes on conventional anatomical imaging techniques, such as computed tomography (CT) and magnetic resonance imaging (MRI), but it cannot differentiate infection from sterile inflammation [7].

Planar scintigraphy or single photon emission computed tomography (SPECT) with autologous radiolabeled white blood cells (WBC) are more specific molecular imaging modalities for diagnosis of infection [5,6]. The main mechanisms of uptake of radiolabeled WBC at the site of infection are cellular migration and target-specific localization: in fact, radiolabeled WBC localize to infected areas via chemotaxis and diapedesis. In vitro labeling of WBC is a rigorous process that requires the following steps: isolation of WBC from the patient’s blood, separation of WBC from plasma, WBC labeling and suspension in plasma, and reinjection of radiolabeled WBC [5].

The relatively limited spatial resolution of SPECT compared to PET has led some researchers to perform the ex-vivo labeling of WBC with ^18^F-FDG for PET/CT imaging of infectious diseases (^18^F-FDG-WBC PET/CT) with initial promising results [3,6,7]. Therefore, we aimed to perform a systematic review and bivariate meta-analysis about the diagnostic performance of ^18^F-FDG-WBC PET or PET/CT in patients with suspicious infectious diseases to add evidence-based data in this setting [8].

## 2. Methods

This systematic review and meta-analysis was written according to the “Preferred Reporting Items for a Systematic Review and Meta-Analysis of Diagnostic Test Accuracy Studies” (PRISMA-DTA statement), a guideline which describes the items required for reporting in systematic reviews and meta-analyses of diagnostic accuracy studies [9].

### 2.1. Search Strategy

The co-authors performed a comprehensive computer literature search of PubMed/MEDLINE, Cochrane library and Embase databases to find relevant published studies on the diagnostic accuracy of ^18^F-FDG-WBC PET or PET/CT in patients with suspicious infectious diseases.

This search algorithm based on a combination of terms was created and used: (A) “FDG” OR “fluorodeoxyglucose” AND (B) “label*” AND (C) “leukocyte*” OR “leucocyte*” OR “white blood” OR “granulocyte*”. No beginning date limit nor language restrictions were used. The literature search was updated until 26 March 2019. References of the retrieved articles were also screened to search for possible additional articles.

### 2.2. Study Selection

Studies assessing the diagnostic performance of ^18^F-FDG-WBC PET or PET/CT in patients with suspicious infectious diseases were eligible for inclusion in the qualitative (systematic review) and quantitative analysis (meta-analysis). The exclusion criteria for the systematic review were: (a) articles not within the field of interest; (b) editorials or letters, review articles, comments, conference proceedings; and (c) case reports. If studies with possible patient data overlap were found, only the article with more complete information was included in the meta-analysis, whereas all of them were included in the systematic review.

All the co-authors independently screened the abstracts of the retrieved articles, applying the predefined inclusion and exclusion criteria. Subsequently, the researchers independently reviewed the full-text of the selected articles to assess their eligibility for inclusion. Disagreements were solved through a consensus meeting among all co-authors.

### 2.3. Data Extraction

For each selected study, information was collected on basic study characteristics (authors, year of publication, country, study design), patient characteristics (type and number of patients evaluated, age and sex ratio), technical details (type of hybrid imaging used, activity used for WBC labeling, WBC labeling efficiency, injected activity, time interval between radiotracer injection and image acquisition, image analysis and other imaging methods performed for comparison), and data on diagnostic accuracy on a per patient-based analysis (including true positive and true negative findings, false positive and false negative findings, sensitivity, specificity, positive and negative predictive values and accuracy).

### 2.4. Quality Assessment

The quality of the studies included in this systematic review was critically appraised using the revised “Quality Assessment of Diagnostic Accuracy Studies” tool (QUADAS-2) [10]. QUADAS-2 includes four domains (patient selection, index test, reference standard, and flow and timing) and each domain was assessed in terms of risk of bias, and the first three domains were also assessed in terms of concerns regarding applicability [10].

### 2.5. Statistical Analysis

Sensitivity and specificity, positive and negative likelihood ratios (LR+ and LR−) and diagnostic odds ratio (DOR) of visual analysis of ^18^F-FDG-WBC PET or PET/CT in patients with suspicious infectious diseases were obtained from individual studies on a per patient-based analysis. A bivariate random-effects model was used for statistical pooling of data about sensitivity and specificity. This statistical approach takes into account any correlation that may exist between sensitivity and specificity [11]. A random-effects model was used for statistical pooling of LR+, LR− and DOR. Pooled data were presented with 95% confidence intervals (95%CI) and displayed using forest plots. Heterogeneity has been estimated by using the I-square index (I^2^) [12]. Publication bias was assessed through the Egger’s test [13].

Statistical analyses were performed using OpenMeta[Analyst]® software (version 0.1503) funded by the Agency for Healthcare Research and Quality (AHRQ) (Rockville, Maryland, United States).

## 3. Results

### 3.1. Literature Search

Literature search results are summarized in Figure 1.

A total of 160 records were identified through the comprehensive computer literature search of PubMed/MEDLINE, Cochrane library and Embase databases. Screening 160 abstracts, 148 records were excluded: 130 because they were not in the field of interest, 11 as editorials, reviews or letters, seven as case reports. Twelve articles were selected and retrieved in full-text. No additional records were found screening the references of these articles, whereas four articles were excluded after the analysis of the full text. Therefore, eight articles were included in the qualitative analysis (systematic review) [14,15,16,17,18,19,20,21]. One article was excluded from the meta-analysis for possible patient data overlap [16]. Overall, seven articles (236 patients with suspicious infectious diseases) were included in the quantitative analysis (meta-analysis) [14,15,17,18,19,20,21]. The characteristics of the studies included in the systematic review are presented in Table 1 and Table 2. Diagnostic accuracy data from these articles are showed in Table 3, whereas the overall quality assessment of the studies is reported in Figure 2.

### 3.2. Qualitative Analysis (Systematic Review)

#### 3.2.1. Basic Study and Patient Characteristics

Screening the selected databases, eight articles evaluating the diagnostic performance of ^18^F-FDG-WBC PET or PET/CT in patients wih suspicious infectious diseases were selected (Table 1) [14,15,16,17,18,19,20,21]. All the selected articles were prospective single-center studies published from 2006 to 2018 by research groups of different countries from Europe, Asia and America. The mean age of patients included in these studies ranged from 41 to 61 years and the percentage of male patients (sex ratio) ranged from 22% to 96%.

#### 3.2.2. Technical Aspects

Technical details about ^18^F-FDG-WBC PET or PET/CT in the included studies are summarized in Table 2. Hybrid PET/CT was performed in 87.5% of the studies, without contrast-enhanced CT in the majority of cases. Mean WBC labeling efficiency ranged from 70% to 81%.

The injected activity and the time interval between injection and PET acquisition were quite different among the studies. The analysis of PET and PET/CT images was performed by using qualitative criteria (visual analysis) in all the studies. Some authors have used visual scores for PET images, comparing the activity in suspicious lesions and in background regions. Additional semi-quantitative criteria, i.e., through the calculation of the maximal standardized uptake values (SUV_max_), were used in 62% of studies. At visual analysis all the areas of focal increased radiotracer uptake greater than the surrounding tissue and not judged as physiological activity were considered to be abnormal. Normal physiologic biodistribution of ^18^F-FDG-WBC includes the reticuloendothelial system (liver, spleen and bone marrow). Minimal radiotracer activity may be detected in brain, myocardium and urinary bladder (due to ^18^F-FDG eluted from radiolabeled WBC). Compared to ^18^F-FDG PET, no physiological activity is usually demonstrated in bowel, kidneys or ureters by using ^18^F-FDG-WBC PET [14,15,16,17,18,19,20,21].

Histopathological/microbiological results and/or clinical/imaging/biochemical work-up and follow-up were used as reference standard in the included studies.

#### 3.2.3. Main Findings

No adverse effects were seen in any of the patients after the ^18^F-FDG-WBC injection [14,15,16,17,18,19,20,21]. Radiation exposure to the staff performing the labeling procedure varied from 4 to 10 µSv for each patient [18]. When measured, the ^18^F-FDG-WBC viability ranged from 97% to >99% [18,21].

In most of the included studies a good diagnostic performance of ^18^F-FDG-WBC PET or PET/CT in patients wih suspicious infection (including musculoskeletal, vascular or soft-tissue infections) was reported (Table 3).

False negative findings of ^18^F-FDG-WBC PET or PET/CT were due to poor host immune reaction, low virulence or chronic infections, vertebral osteomyelitis, infections in site of physiological radiotracer uptake, prior antibiotic therapy or immunosuppressant drugs administration. False positive findings of ^18^F-FDG-WBC PET or PET/CT were due to hemorrhagic lesions, some cases of aseptic/sterile inflammation, recent surgery or eluition of ^18^F-FDG from radiolabeled WBC over time [14,15,16,17,18,19,20,21].

Compared to conventional imaging methods (CT and MRI) and ^18^F-FDG PET/CT, a higher diagnostic performance was found using ^18^F-FDG-WBC PET or PET/CT in patients with suspicious infection [14,15,16,17,19]. One study found comparable results in terms of diagnostic accuracy among ^18^F-FDG-WBC PET and scintigraphy using WBC labeled with Indium-111-oxine (^111^In-WBC) [21].

### 3.3. Quantitative Analysis (Meta-Analysis)

Seven prospective studies including 236 patients with suspicious infectious diseases were selected for the bivariate meta-analysis [14,15,17,18,19,20,21]. Results of the meta-analysis are presented in Figure 3, Figure 4 and Figure 5.

The sensitivity of ^18^F-FDG-WBC PET or PET/CT in patients with suspicious infectious diseases ranged from 64.3.8% to 100%, with a pooled estimate of 86.3% (95%CI: 75–92.9%). The specificity of ^18^F-FDG-WBC PET or PET/CT in patients with suspicious infectious diseases ranged from 60% to 100%, with a pooled estimate of 92% (95%CI: 79.8–97.1%). The pooled LR+, LR− and DOR were 6.6 (95%CI: 3.1–14.1), 0.2 (95%CI: 0.12–0.33), and 43.5 (95%CI: 12.2–155), respectively (Figure 3, Figure 4 and Figure 5). No significant statistical heterogeneity among the studies was found for all the metrics evaluated (*I*^2^ < 50%). No significant publication bias was detected by the Egger’s test (*p* > 0.1).

## 4. Discussion

Osman and Danpure first described the in vitro labeling of WBC with ^18^F-FDG, showing that 87% of ^18^F-FDG is associated with labeling of granulocytes [22]. Subsequently, several studies have evaluated the diagnostic performance of ^18^F-FDG-WBC PET or PET/CT in different types of infectious diseases [14,15,16,17,18,19,20,21], but these studies have limited power due to the small number of patients enrolled. Therefore, we have pooled data reported in the published studies through a bivariate meta-analysis to derive more robust estimates on the diagnostic performance of ^18^F-FDG-WBC PET or PET/CT in this setting.

Overall, despite the relatively limited data, our systematic review and bivariate meta-analysis demonstrated a good diagnostic performance of ^18^F-FDG-WBC PET or PET/CT in patients with suspicious infectious diseases without significant adverse effects. To this regard, possible false negative findings (i.e., due to poor host immune reaction, low virulence or chronic pattern of disease, vertebral osteomyelitis, infections in site of physiological radiotracer uptake, prior antibiotic therapy or immunosuppressant drugs administration) and false positive findings (i.e., due to hemorrhagic lesions, some cases of aseptic/sterile inflammation, recent surgery or eluition of ^18^F-FDG from radiolabeled WBC over time) should be taken into account when nuclear medicine physicians interpret ^18^F-FDG-WBC PET or PET/CT images [14,15,16,17,18,19,20,21].

About the comparison of ^18^F-FDG-WBC PET or PET/CT with other imaging modalities in patients with suspicious infectious diseases, a higher diagnostic performance was found using ^18^F-FDG-WBC PET/CT compared to conventional imaging methods (CT or MRI) and ^18^F-FDG PET/CT [14,15,16,17,19].

The normal biodistribution of ^18^F-FDG includes the brain and the genitourinary tract and variably high activity in the myocardium, bone marrow, gastro-intestinal tract and liver. ^18^F-FDG PET/CT may have a reduced performance for the detection of infection in these sites [7]. Conversely, the physiological uptake of ^18^F-FDG-WBC essentially occurs within the reticuloendothelial system. The absence of gastrointestinal and renal uptake and the faint brain and myocardial uptake makes ^18^F-FDG-WBC PET/CT a suitable imaging method for the assessment of intra-abdominal, renal, cardiac and cerebral infectious diseases [14,15,16,17,18,19,20,21]. Another difference between ^18^F-FDG-WBC PET/CT and ^18^F-FDG PET/CT relates to the different cellular types involved in the signal detected in infected sites. ^18^F-FDG reveals different types of inflammatory cells, including macrophages, whereas ^18^F-FDG-WBC essentially reveals active diapedesis of granulocytes through chemotactic processes [20,23], and this could explain the higher diagnostic performance obtained by using ^18^F-FDG-WBC PET/CT compared to ^18^F-FDG PET/CT in patients with suspicious infectious diseases [17,19].

Only one study compared ^18^F-FDG-WBC PET and ^111^In-WBC scintigraphy in patients with suspicious infectious diseases reporting a similar diagnostic performance of these imaging methods [21]. However, further considerations should be added to this regard. The labeling efficiency of ^18^F-FDG-WBC is variable and significantly lower than that of WBC radiolabeled with gamma-emitting tracers [15,17,18,19,20,21]; although not examined in detail, the leukocyte glucose transporter expression, the serum glucose levels, and the presence of intrinsic proteins have been thought to affect the labeling efficiency of ^18^F-FDG-WBC [3]. Nevertheless, the relative low labeling efficiency seems to not affect the diagnostic accuracy of ^18^F-FDG-WBC PET or PET/CT in detecting infectious diseases. On the other hand, the mean cell viability of ^18^F-FDG-WBC was very high and comparable to that of radiolabelled WBC used for scintigraphy [18,21]; thus, the ^18^F-FDG labeling procedure does not affect the WBC viability. Another factor that should be considered is the short physical half-life of ^18^F which excludes imaging at 24 h after radiotracer injection (as usually performed for radiolabeled WBC scintigraphy); the WBC labeling and loading time may also have an impact due to the fast radiotracer decay. Some advantages of ^18^F-FDG-WBC PET/CT compared to radiolabelled WBC scintigraphy or SPECT are the better image quality and resolution and the reduced imaging time [21]. Dosimetry of ^18^F-FDG-WBC PET for activities of 225–315 MBq was found to be comparable to results with ^111^In-WBC scintigraphy [24].

Although the short half-life of ^18^F excludes imaging at 24 h after radiotracer injection (which is known to increase the specificity of radiolabeled WBC scintigraphy), a high sensitivity and specificity for infection was found in our pooled analysis on ^18^F-FDG-WBC PET or PET/CT; the better resolution of PET imaging compared to planar scintigraphy and SPECT may have contributed to the high sensitivity and the high level of anatomic detail provided by the co-registered CT images may have contributed to the high specificity [20].

Contrast enhancement could further improve the diagnostic performance of ^18^F-FDG-WBC PET/CT for detection of infection, in particular for visceral localizations, but this should be better evaluated in further studies [18,20]. Moreover, the use of semi-quantitative criteria of interpretation of ^18^F-FDG-WBC PET/CT (by using SUVmax thresholds) seems to be not sufficiently accurate to make a differential diagnosis between infectious and non-infectious conditions, but semi-quantitative analysis can be used as an adjunct tool to visual analysis for PET interpretation, even if a possible effect of cell division on the SUVmax within the imaging time frame cannot be excluded [17].

Different imaging times after ^18^F-FDG-WBC injection were adopted in the selected studies [14,15,16,17,18,19,20,21]: delayed image acquisition may improve the contrast between infectious foci and background, but might increase the possibility of ^18^F-FDG decay and detachment from WBCs [16]; additional studies are needed to identify the optimal ^18^F-FDG-WBC PET/CT imaging protocol.

Diagnostic accuracy of an imaging test is not a measure of clinical effectiveness and high diagnostic performance does not necessarily result in improved patient outcomes. Other factors beyond the diagnostic performance should influence the choice of an imaging modality in patients with suspicious infectious disease (i.e., availability, radiation dose, safety, examination time, legal, organization, economic aspects). Overall, our systematic review and meta-analysis demonstrated a good diagnostic performance of ^18^F-FDG-WBC PET/CT in patients with suspicious infectious diseases, but larger prospective and multicenter studies on this topic are needed, in particular, more comparison studies with radiolabeled WBC scintigraphy are warranted. There are some drawbacks of ^18^F-FDG-WBC PET/CT that should be underlined, including the relatively long time for WBC labeling, the need for high ^18^F-FDG activities for labeling, variable labeling efficiency, risk of contamination by direct contact with blood products and radiation exposure [3,6,7,19]. Furthermore, the possibility of an increased risk of lymphoid malignancies associated with the administration of ^18^F-FDG-WBC is a very controversial subject that needs further investigation [25]. Lastly, there are still insufficient in vitro data on putative elution of ^18^F-FDG from radiolabeled WBC over time [7].

Some limitations and biases of our meta-analysis should be taken into account. First of all, a limited number of studies were available for the meta-analysis. Moreover, as a composite reference standard was used in most of the studies a possible verification bias could not be excluded. Furthermore, based on the information provided in the studies selected for the meta analysis, a selection bias could be present. Heterogeneity among studies (i.e., due to baseline differences among the patients included, diversity in methodological aspects, and different study quality) may represent a potential source of bias in a meta-analysis [11]. We have not detected a statistically significant heterogeneity among the included studies in our meta-analysis, although the significant differences about the patient population evaluated and the technical details in the included studies.

## 5. Conclusions

Based on available literature data, ^18^F-FDG-WBC PET or PET/CT seem to demonstrate a good diagnostic performance in detecting infectious diseases.

The literature on this topic is still limited and further investigations on ^18^F-FDG-WBC PET/CT in patients with suspicious infectious diseases are warranted. Based on available data, this imaging method should not be considered as standard clinical practice, but it could be used in a research setting.

## Figures and Tables

**Figure 1 diagnostics-09-00060-f001:**
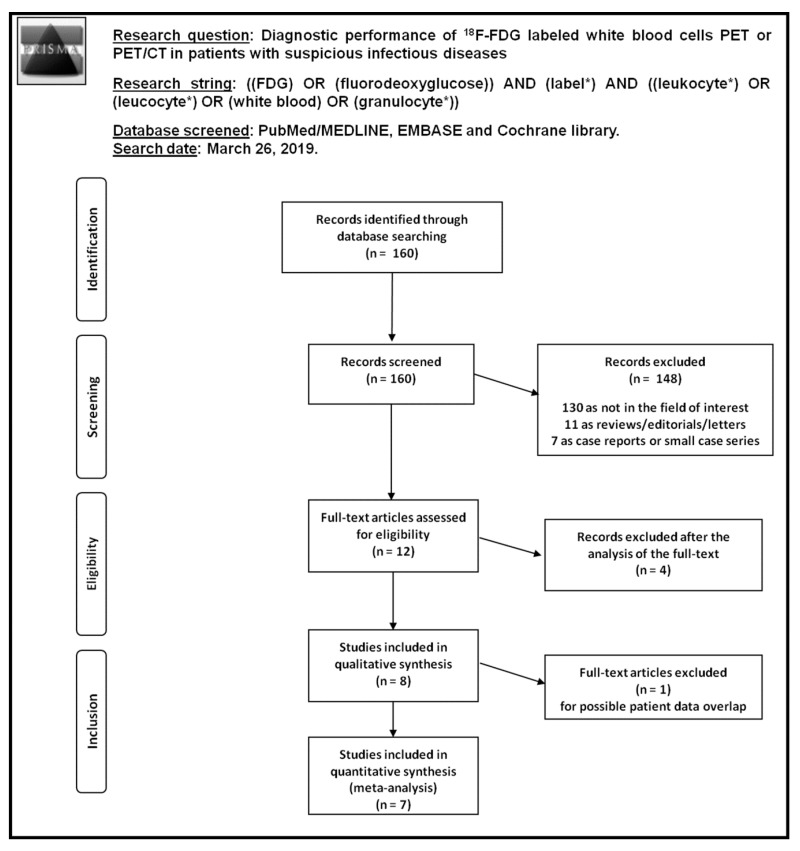
Flow chart of the search for eligible studies on the diagnostic performance of fluorine-18-fluorodeoxyglucose (^18^F-FDG-WBC PET or PET/CT) in patients with suspicious infectious diseases.

**Figure 2 diagnostics-09-00060-f002:**
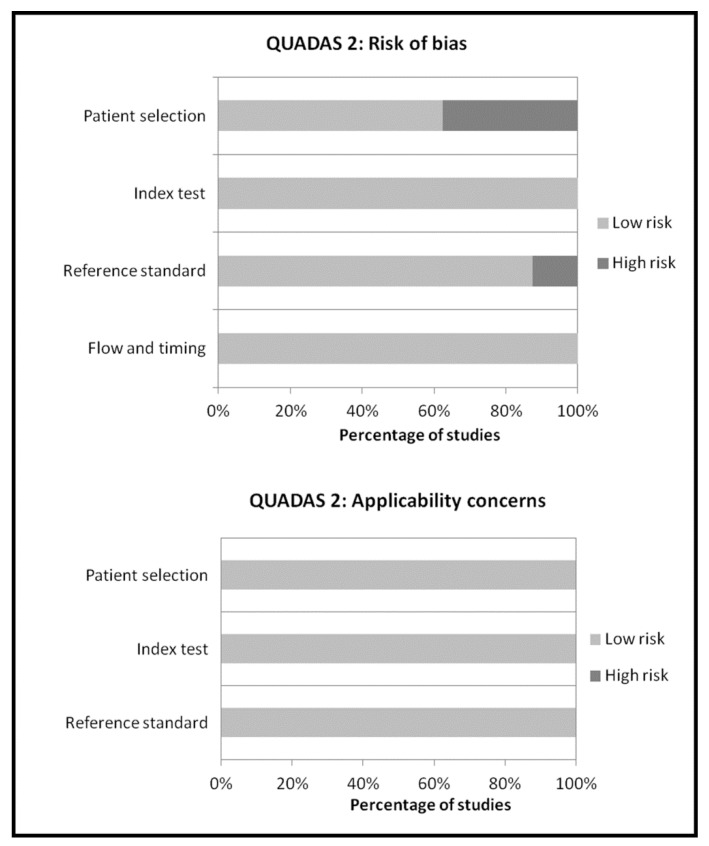
Overall quality assessment of the studies included in the systematic review according to Quality Assessment of Diagnostic Accuracy Studies (QUADAS-2) tool.

**Figure 3 diagnostics-09-00060-f003:**
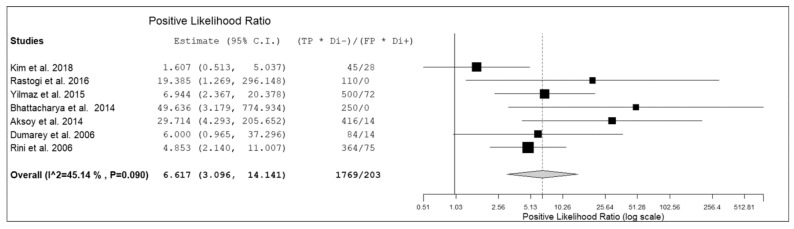
Forest plot of individual studies and pooled positive likelihood ratio of ^18^F-FDG-WBC PET or PET/CT in patients with suspicious infectious diseases, including 95% confidence interval (95% CI). The size of the squares indicates the weight of each study.

**Figure 4 diagnostics-09-00060-f004:**
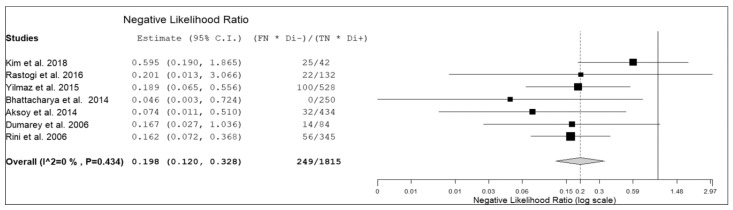
Forest plot of individual studies and pooled negative likelihood ratio of ^18^F-FDG-WBC PET or PET/CT in patients with suspicious infectious diseases, including 95% confidence interval (95%CI). The size of the squares indicates the weight of each study.

**Figure 5 diagnostics-09-00060-f005:**
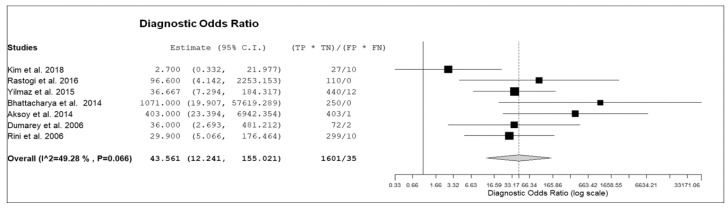
Forest plot of individual studies and pooled diagnostic odds ratio of ^18^F-FDG-WBC PET or PET/CT in patients with suspicious infectious diseases, including 95% confidence interval (95%CI). The size of the squares indicates the weight of each study.

**Table 1 diagnostics-09-00060-t001:** Basic study and patient characteristics.

Authors	Year	Country	Study Design	Type of Patients Evaluated	No. of Patients Performing ^18^F-FDG-WBC PET or PET/CT	Mean Age (Years)	% Male
Kim et al. [14]	2018	South Korea	prospective single center	Patients with autosomal dominant polycystic kidney disease and suspicious cyst infection	19	54 (38–73)	37%
Rastogi et al. [15]	2016	India	prospective single center	Patients with Charcot’s neuroarthropathy and suspicious diabetic foot osteomyelitis	23	57.9 ± 7.8	96%
Kwon et al. [16]	2016	South Korea	prospective single center	Patients with autosomal dominant polycystic kidney disease and suspicious cyst infection	17	53 (38–73)	35%
Yilmaz et al. [17]	2015	Turkey	prospective single center	Patients with suspicious musculoskeletal, vascular or soft-tissue infection	49	55.7 (16–89)	55%
Bhattacharya et al. [18]	2014	India	prospective single center	Patients with suspicious infected fluid collections in acute pancreatitis	41	41 ± 11.5	68%
Aksoy et al. [19]	2014	Turkey	prospective single center	Patients with suspicious prosthetic joint infection	46	61 ± 12.2	22%
Dumarey et al. [20]	2006	Belgium	prospective single center	Patients with suspicious infectious diseases	21	56 (24–84)	62%
Rini et al. [21]	2006	USA	prospective single center	Patients with suspicious infectious diseases	43	59 (32–86)	47%

Legend: ^18^F-FDG-WBC = white blood cells labeled with fluorine-18 fluorodeoxyglucose; CT = computed tomography; NA = not available; PET = positron emission tomography.

**Table 2 diagnostics-09-00060-t002:** Technical aspects of the included studies.

Authors	Hybrid Imaging Modality	Activity Used for WBC Labeling	Mean WBC Labeling Efficiency (%)	Injected Activity	Image Analysis	Other Imaging Modalities Performed for Comparison
Kim et al. [14]	PET/CT (low-dose CT)	740 MBq	NR	NR	visual	MRI or CT
Rastogi et al. [15]	PET/CT (low-dose CT)	NR	73 ± 28	NR	visual and semi-quantitative (SUV_max_)	^18^F-fluoride PET/CT and MRI
Kwon et al. [16]	PET/CT (low-dose CT)	740 MBq	NR	189 ± 107 MBq	visual	MRI or CT
Yilmaz et al. [17]	PET/CT (low-dose CT)	740–925 MBq	70 ± 13	70-433 MBq	visual and semi-quantitative (SUV_max_)	^18^F-FDG PET/CT
Bhattacharya et al. [18]	PET/CT (low-dose or contrast enhanced CT)	314.5–555 MBq	81 ± 17	262.7 ± 70.3 MBq	visual and semi-quantitative (SUV_max_)	-
Aksoy et al. [19]	PET/CT (low-dose CT)	740–925 MBq	75 ± 17	296–703 MBq	visual and semi-quantitative (SUV_max_)	^18^F-FDG PET/CT
Dumarey et al. [20]	PET/CT (low-dose CT)	740 MBq	75 ± 21	303 ± 111 MBq	visual and semi-quantitative (SUV_max_)	-
Rini et al. [21]	No (PET only)	660–740 MBq	72 ± 8	196–315 MBq	visual	^111^In-WBC scintigraphy

Legend: ^18^F-FDG = Fluorine-18 Fluorodeoxyglucose; CT = computed tomography; ^111^In = Indium-111; MBq = MegaBecquerel; min = minutes; MRI = magnetic resonance imaging; NR = not reported; PET/CT = positron emission tomography/computed tomography; SUV_max_ = maximal standardized uptake value; WBC = white blood cells.

**Table 3 diagnostics-09-00060-t003:** Diagnostic accuracy data of ^18^F-FDG WBC PET or PET/CT in the included studies on a per patient-based analysis.

Authors	True Positive	False Negative	False Positive	True Negative	Sensitivity	Specificity	PPV	NPV	Accuracy
Kim et al. [14]	9	5	2	3	64.3%	60%	81.8%	37.5%	63.2%
Rastogi et al. [15]	10	2	0	11	83.3%	100%	100%	84.6%	91.3%
Kwon et al. [16] *	NA	NA	NA	NA	85.7%	87.5%	85.7%	87.5%	NA
Yilmaz et al. [17]	20	4	3	22	83.3%	88%	87%	84.6%	85.7%
Bhattacharya et al. [18]	10	0	0	25	100%	100%	100%	100%	100%
Aksoy et al. [19]	13	1	1	31	92.9%	96.9%	92.9%	96.9%	95.7%
Dumarey et al. [20]	12	2	1	6	85.7%	85.7%	92.3%	75%	85.7%
Rini et al. [21]	13	2	5	23	86.7%	82.1%	72.2%	92%	83.7%

Legend: NA = not available; * = excluded from the meta-analysis for possible data overlap; NPV = negative predictive value; PPV = positive predictive value.

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
