# Peer review of "Diagnostic Performance of PET or PET/CT Using ^18^F-FDG Labeled White Blood Cells in Infectious Diseases: A Systematic Review and a Bivariate Meta-Analysis"

_diagnostics, 2019, doi:10.3390/diagnostics9020060_

Round 1
Reviewer 1 Report
The authors describe the detection of sites of infection in patients using pre- labeled WBCs with 19F-FDG using PET imaging.
1)The authors should discuss the patient bias more throughly based on the details provided in the manuscripts selected for this meta analysis. It seems that even in the small cohorts there seems to be selection bias.
2) Table 2 has a column for other imaging modalities. Yilmaz et al has 19F-FDG PET/CT listed. Does this mean a direct systemic administration of the imaging agent instead of injection of labeled cells that are fed the imaging agent in-vitro (WBCs)?
3)The authors should discuss cell division and its associated effect on the SUV
within the imaging time frame?
4)Table 2 has a column called time interval between radiotracer injection and image acquisition is misleading since the WBCs are loaded with FDG prior to injection. So infact, the cells carrying the radiotracer are injected back. Authors should discuss the impact of cell loading time as this time will also have decay of the radiotracer.
5) Abstract and introduction should be checked for grammatical errors.
This is a useful report.
Author Response
Response to Reviewer #1:
The authors describe the detection of sites of infection in patients using pre-labeled WBCs with 18F-FDG using PET imaging.
- The authors should discuss the patient bias more throughly based on the details provided in the manuscripts selected for this meta analysis. It seems that even in the small cohorts there seems to be selection bias.
Response: according to the Reviewer’s comment we have added this statement in the discussion of the revised manuscript: “Furthermore, based on the information provided in the studies selected for the meta analysis, a selection bias could be present.”
- Table 2 has a column for other imaging modalities. Yilmaz et al has 18F-FDG PET/CT listed. Does this mean a direct systemic administration of the imaging agent instead of injection of labeled cells that are fed the imaging agent in-vitro (WBCs)?
Response: in the study of Yilmaz et al. the patients underwent both 18F-FDG PET/CT and 18F-FDG-WBC PET/CT and these imaging methods were compared.
- The authors should discuss cell division and its associated effect on the SUV within the imaging time frame.
Response: according to the Reviewer’s comment we have added this statement in the discussion “a possible effect of cell division on the SUVmax within the imaging time frame cannot be excluded”.
- Table 2 has a column called time interval between radiotracer injection and image acquisition is misleading since the WBCs are loaded with FDG prior to injection. So, the cells carrying the radiotracer are injected back. Authors should discuss the impact of cell loading time as this time will also have decay of the radiotracer.Response: We have deleted the column “time interval between radiotracer injection and image acquisition” from Table 2 because misleading. According to the Reviewer’s comment we have added this statement in the discussion “the WBC labeling time may also have an impact due to the fast radiotracer decay”.
- Abstract and introduction should be checked for grammatical errors.
Response: We have checked abstract and introduction for grammatical errors, performing some changes in these sections.
Reviewer 2 Report
Meyer, et al., present a systematic review study on the performance of PET or PET/CT using 18F FDG labeled white blood cells for the diagnosis of infectious diseases. The analysis is based on a bivariate meta-analysis of published literature from different biomedical database. The authors created a statistical method to quantitatively analyze the clinical data published in these literature. However, the significance of this manuscript was compromised by the small pool of publication reviewed in this paper. Meanwhile, I have major concern about the statistical analysis of accuracy data from different studies using different pool of patients. When the sample pool size is different, it can be challenging to cross-analyze the data from different pool.
Minor points: Figures need to be remade into high resolution figures. In particular, Figure 3 and 4 is too vague to judge the significance of these data.
Author Response
- Meyer et al. present a systematic review study on the performance of PET or PET/CT using 18F FDG labeled white blood cells for the diagnosis of infectious diseases. The analysis is based on a bivariate meta-analysis of published literature from different biomedical database. The authors created a statistical method to quantitatively analyze the clinical data published in these literature. However, the significance of this manuscript was compromised by the small pool of publication reviewed in this paper.
Response: the limited number of publications included is a limitation of our analysis as we have underlined in the discussion. Nevertheless, this small number has not hampered the statistical analysis. As we have found a limited number of articles through the systematic review process, we have reported in the conclusions that “The literature on this topic is still limited and further investigations on 18F-FDG-WBC PET/CT in patients with suspicious infectious diseases are warranted.”
We have added in the conclusion of the revised manuscript that: “Based on available data, this imaging method should not be considered as standard clinical practice, but it could be used in a research setting.”
- Meanwhile, I have major concern about the statistical analysis of accuracy data from different studies using different pool of patients. When the sample pool size is different, it can be challenging to cross-analyze the data from different pool.
Response: We have not detected a statistically significant heterogeneity among the included studies in our meta-analysis, based on the calculation of the I2 index, although the significant differences about the patient population evaluated and the technical details in the included studies.
- Minor points: Figures need to be remade into high resolution figures. In particular, Figure 3 and 4 is too vague to judge the significance of these data.
Response: We have deleted Figure 3 and divided Figure 4 in three different figures. The graphs are provided in this format by the statistical software.
Round 2
Reviewer 2 Report
The manuscript has been improved.